

# Analysis of synonymous codon usage patterns in sixty-four different bivalve species

Marco Gerdol[1], Gianluca De Moro[1], Paola Venier[2] and
Alberto Pallavicini[1]

[1] Department of Life Sciences, University of Trieste, Trieste, Italy
[2] Department of Biology, University of Padova, Padova, Italy

## ABSTRACT

Synonymous codon usage bias (CUB) is a defined as the non-random usage of codons encoding the same amino acid across different genomes. This phenomenon is common to all organisms and the real weight of the many factors involved in its shaping still remains to be fully determined. So far, relatively little attention has been put in the analysis of CUB in bivalve mollusks due to the limited genomic data available. Taking advantage of the massive sequence data generated from next generation sequencing projects, we explored codon preferences in 64 different species pertaining to the six major evolutionary lineages in Bivalvia. We detected remarkable differences across species, which are only partially dependent on phylogeny. While the intensity of CUB is mild in most organisms, a heterogeneous group of species (including Arcida and Mytilida, among the others) display higher bias and a strong preference for AT-ending codons. We show that the relative strength and direction of mutational bias, selection for translational efficiency and for translational accuracy contribute to the establishment of synonymous codon usage in bivalves. Although many aspects underlying bivalve CUB still remain obscure, we provide for the first time an overview of this phenomenon in this large, commercially and environmentally important, class of marine invertebrates.

## INTRODUCTION

Codon usage bias (CUB), intended as the non-random usage of synonymous codons in the protein translation process, can be observed in virtually all organisms. This phenomenon widely varies across different species and it is expected to significantly influence molecular genome evolution (*Hershberg & Petrov, 2008*; *Sharp, Emery & Zeng, 2010*; *Plotkin & Kudla, 2011*).

The mechanisms behind CUB are complex and not completely understood, since a large number of different intertwined biological factors are correlated with the choice of optimal codons. These include the GC content, both at gene and at whole genome level (*Sueoka & Kawanishi, 2000*; *Zeeberg, 2002*; *Wan et al., 2004*; *Palidwor, Perkins & Xia, 2010*),

Corresponding author
Marco Gerdol, mgerdol@units.it

gene length, structure, expression levels and transcriptional efficiency (*Gouy & Gautier, 1982*; *Sharp, Tuohy & Mosurski, 1986*; *Bains, 1987*; *Eyre-Walker, 1996*; *Duret & Mouchiroud, 1999*), protein structure and amino acid composition (*D'Onofrio et al., 1991*; *Xie et al., 1998*), tRNA abundance (*Ikemura, 1985*), selection, mutational bias and random drift (*Bulmer, 1991*; *Sharp et al., 1993*; *Kliman & Hey, 1994*).

While CUB has been extensively studied in many viruses, prokaryotes, as well as in a number of eukaryote model species (*Stenico, Lloyd & Sharp, 1994*; *Powell & Moriyama, 1997*; *Ermolaeva, 2001*; *Jenkins & Holmes, 2003*; *Gu et al., 2004*; *Mitreva et al., 2006*; *Vicario, Moriyama & Powell, 2007*; *Behura & Severson, 2012*), so far little attention has been focused on non-model invertebrates. In particular, the large phylum of Mollusca has been almost completely neglected, even though it comprises more than 9,000 species including some of a great relevance as sea food and as sentinel organisms for coastal water biomonitoring (*Gosling, 2003*).

For a long time, genomic studies in Bivalvia have been limited by the lack of sequence data. However, the recent advances in the field of high throughput sequencing permitted to unravel the genomes of *Crassostrea gigas* and *Pinctada fucata* (*Takeuchi et al., 2012*; *Zhang et al., 2012*) and to obtain a massive amount of transcriptomic data, useful for large-scale comparative studies (*Suárez-Ulloa et al., 2013*).

The only study performed so far on codon usage in Bivalvia was based on ESTs generated by Sanger sequencing and targeted a single species, the Pacific oyster *C. gigas* (*Sauvage et al., 2007*). Here, we provide the first comprehensive study of CUB in bivalves: based on the analysis of 2,846 evolutionarily conserved protein-coding genes in 64 different species, we calculated codon frequencies and Relative Synonymous Codon Usage (RSCU) values for each species, thus identifying both preferred and avoided codons, and calculating the overall CUB at the species transcriptome level.

Our data highlight significant differences among the analyzed species and clearly identify a bivalve subgroup with an increased codon bias, comprising Mytilida, Arcida and several different species of the Imparidentia lineage. We discuss the evolution of CUB in Bivalvia in relation with the possible underlying factors such as species phylogeny, mutational bias and natural selection. Overall, the results of these analyses bring new insights on the evolution of bivalve genomes and on the major forces driving the evolution of codon usage in bivalves and will provide a reference for improving the annotation of protein-coding genes in future bivalve genome sequencing efforts.

## MATERIALS AND METHODS

### Data sources

We considered two bivalve mollusk species with a fully sequenced genome (*C. gigas* and *P. fucata*) (*Takeuchi et al., 2012*; *Zhang et al., 2012*) and 62 other species whose transcriptome has been sequenced using next generation sequencing technologies and deposited in public sequence databases. When both 454 Life Sciences and Illumina-generated sequencing reads were available for a same species, the latter were chosen due to higher throughput and lower rate of sequencing errors. Namely, Illumina reads were used for species *Anadara*

*trapezia, Arctica islandica, Argopecten irradians, Astarte sulcata, Atrina rigida, Azumapecten farreri, Bathymodiolus platifrons, Cardites antiquata, Cerastoderma edule, Corbicula fluminea, Crassostrea angulata, Crassostrea corteziensis, Crassostrea hongkongensis, Crassostrea virginica, Cycladicama cumingii, Cyrenoida floridana, Diplodonta* sp. VG-2014, *Donacilla cornea, Elliptio complanata, Ennucula tenuis, Eucrassatella cumingii, Galeomma turtoni, Glossus humanus, Hiatella arctica, Lampsilis cardium, Lamychaena hians, Mactra chinensis, Margaritifera margaritifera, Mercenaria campechiensis, Meretrix meretrix, Mizuhopecten yessoensis, Mya arenaria, Myochama anomioides, Mytilus californianus, Mytilus edulis, Mytilus galloprovincialis, Mytilus trossulus, Neotrigonia margaritacea, Ostrea chilensis, Ostrea edulis, Ostrea lurida, Ostreola stentina, Pecten maximus, Perna viridis, Pinctada martensii, Placopecten magellanicus, Polymesoda caroliniana, Pyganodon grandis, Ruditapes decussatus, Ruditapes philippinarum, Sinonovacula constricta, Solemya velum, Sphaerium nucleus, Uniomerus tetralasmus* and *Villosa lienosa* (Qin et al., 2012; Ghiselli et al., 2012; Chen et al., 2013; Meng et al., 2013; Gerdol et al., 2014; Fu et al., 2014; Zhang et al., 2014; Pauletto et al., 2014; De Sousa et al., 2014; Zhao et al., 2014; Cornman et al., 2014; Zapata et al., 2014; Prentis & Pavasovic, 2014; González et al., 2015). The 454 Life Sciences sequences were used for *Bathymodiolus azoricus, Geukensia demissa, Laternula elliptica, Mimachlamys nobilis, Pinctada maxima, Saccostrea glomerata*, and *Tegillarca granosa* (Clark et al., 2010; Philipp et al., 2012; Egas et al., 2012; Jones et al., 2013; Fields et al., 2014). Details about the data used for the different species are provided in Table S1.

Sequence data were processed as follows: predicted CDS from the fully sequenced genomes of *C. gigas* (release 9) and *P. fucata* were retrieved from http://oysterdb.cn and http://marinegenomics.oist.jp/pinctada_fucata, respectively. *De novo* transcriptome assemblies were performed for all the other 62 bivalve species with the CLC Genomics Workbench (v.7.5, CLC Bio, Aarhus, Denmark) using the *de novo assembly* tool with "automatic word size" and "automatic bubble size" parameters selected, and setting the minimum allowed contig length to 300 bp.

In all transcriptomes, ORFs (Open Reading Frames) longer than 100 codons were predicted with TransDecoder (http://transdecoder.sourceforge.net). We selected the predicted CDS of *C. gigas*, and of one representative species for the Imparidentia (*R. decussatus*), Protobranchia (*S. velum*) and Palaeoheterodonta (*P. grandis*) lineages to identify a subset of evolutionarily conserved protein-coding genes with a 1:1 orthology ratio across Bivalvia. This was achieved by performing reciprocal tBLASTx searches (the $e$-value threshold was set a $1 \times 10^{-10}$ and only hits displaying sequence identity >50% were considered). This procedure resulted in a selection of 2,846 conserved protein-coding genes, whose orthologous sequences were retrieved in the remaining 60 species. Due to the heterogeneous tissue and developmental stage origin, the different sequencing platforms and depth applied, several of these evolutionarily conserved sequences could not be identified or were fragmented in some transcriptomes. In order to ensure a minimum quality criteria, all the selected species had to display at least 25% of the sequences included in the dataset of evolutionarily conserved genes, with an average length >500 nucleotides.

A number of additional transcriptomes derived from publicly available data did not meet such criteria and were therefore not included in our analyses (Table S2).

## Codon frequencies and codon usage statistics

The sets of evolutionarily conserved genes retrieved for each species were individually processed with the *cusp* tool of the EMBOSS package (*Rice, Longden & Bleasby, 2000*) obtaining codon frequencies and GC composition for each codon position. RSCU values for each individual codon were calculated for each species as described by Sharp and colleagues (*1986*). The effective number of codons (ENC) for each species was calculated according to *Wright (1990)* using the EMBOSS *chips* tool, summing codons over al sequences (*Rice, Longden & Bleasby, 2000*). The sENC-X values were determined for every amino acid for each species and scaled to a range of values between 0 and 1 according to *Moriyama & Powell (1997)*. EMBOSS *chips* was also used to calculate ENC for individual genes whenever necessary. We identified a reference set of 50 highly expressed genes for the calculation of Codon Adaptation Index (CAI) based on the average expression in *C. gigas* digestive gland (SRA:SRX093412), gills (SRA:SRX093414) and hemocytes (SRA:SRX093417) RNA-seq libraries and their inclusion in the above mentioned set of 2,846 genes conserved across bivalves. Gene expression was calculated as TPM (Transcripts Per Million) (*Wagner, Kin & Lynch, 2012*), with the RNA-seq mapping tool included in the CLC Genomics Workbench 8.5 (Aarhus, Denmark), setting length and similarity fraction parameters to 0.75 and 0.98 and insertion/deletion/mismatch penalties to 3. Orthologous genes were used for CAI calculation in other species. CAI values were computed with CAI calculator 2 (*Wu, Culley & Zhang, 2005*). The gene expression levels of *M. galloprovincialis* transcripts were calculated using the digestive gland (SRA:SRX126945-8), gills (SRA:SRX389466) and hemocytes (SRA:SRX389338) RNA-seq libraries (*Gerdol et al., 2014*; *Moreira et al., 2015*).

Scatter plots were generated between ENC and the average GC content calculated at the third codon position ($GC_3$) for each species, between ENC and sENCx and between ENC and CAI; Paerson correlation coefficients and linear regression analyses were computed with R 3.1.0 (http://www.r-project.org).

## Hierarchical clustering analysis

The Relative Synonymous Codon Usage (RSCU) values calculated for the 59 informative codons in each species were used to build a tabular file. STOP, ATG (encoding Met) and TGG (encoding Trp) codons were excluded from this analysis. This file was used as an input for Cluster 3 (*De Hoon et al., 2004*), thus generating a species distance matrix. Hierarchical clustering was performed by using the Euclidean distance as a similarity metric and complete linkage as a clustering method.

# RESULTS AND DISCUSSION

## CUB varies across bivalve species

The codon frequencies and RSCU values calculated for the 64 bivalve species analyzed are displayed in Tables S3 and S4. As shown in the comparative overview of Fig. 1, RSCU values

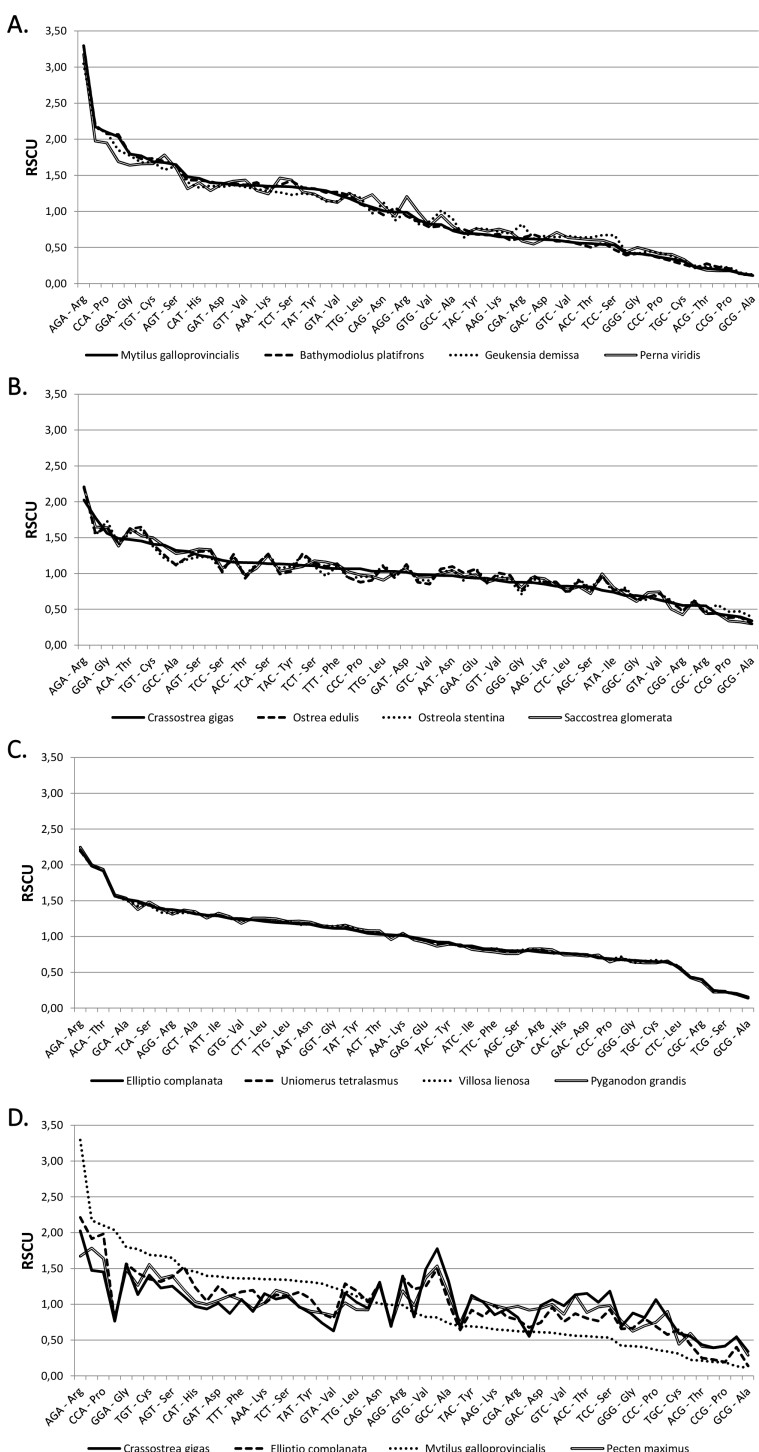

**Figure 1  Relative synonymous codon usage across bivalves.** RSCU values (*Y* axis) in four Mytilida (A), Ostreoidea (B) and Unionida (C) species. (D) shows a comparison between representative species from the three above mentioned orders and the Pectinida *Pecten maximus*. Codons are ordered by decreasing RSCU value on the *X* axis, based on *Mytilus galloprovincialis* (A and D), *Crassostrea gigas* (B) and *Elliptio complanata* (C).

are usually very similar in closely related species, such as in the case of Mytilida, Ostreoidea and Unionida (A, B and C), but marked differences can be observed in an higher-order comparison (D).

Different species clearly show a different tendency to the preferential usage of specific codons, as exemplified by the average Effective Number of Codons (ENC) (*Wright, 1990*) values in Table 1. Overall, the observed ENC values range between 40.65 (in the Chinese surf clam *M. chinensis*) and 56.80 (*G. turtoni*) across the analyzed species, while the theoretical value is comprised between 21 (if only a single codon is used for each amino acid) and 61 (if all codons are used with equal frequency). Most bivalve species display a weak CUB, using on average over 50 out of the 61 available codons, and only a limited number of bivalve species display an ENC value comparable to that of other invertebrates (the ENC range is 45–48 in *Drosophila* and nematodes) (*Powell & Moriyama, 1997*; *Mitreva et al., 2006*; *Vicario, Moriyama & Powell, 2007*).

## Species clustering based on CUB does not reflect the evolutionary history of Bivalvia

We computed RSCU values for the 59 informative codons of each species to perform a hierarchical clustering of species with Cluster 3, thereby investigating the role of CUB in the evolution of bivalve genomes. The resulting dendrogram is shown in Fig. 2. Although phylogeny and CUB-based clustering are in agreement, in several cases, up to the order level, the six major lineages of Anomalodesmata, Archiheterodonta, Imparidentia, Palaeoheterodonta, Protobranchia and Pteriomorphia expected from molecular phylogeny (*Bieler et al., 2014*) are hardly distinguishable. This observation is consistent with data previously reported for nematodes by Mitreva and colleagues (*2006*), who identified a connection between codon distribution and phylogeny only for closely related species, up to the genus level. In bivalves such a relationship seems to extend a bit further, in some cases up to one of the six major evolutionary lineages, as for instance in Palaeoheterodontha, which include the freshwater mussels of the order Unionida and the saltwater clams of the order Trigoniida, or Archiheterodonta, which only comprise four relatively small extant families. While in some cases (e.g., Mytilida and Ostreoidea) all the species maintain a similar usage of synonymous codons (Fig. 1), in others (e.g., Venerida) remarkable differences among species are clearly visible.

In essence, the clustering based on codon usage divides the bivalve species into two largely divergent groups:

(I) The first group is very heterogeneous, comprising 43 species with ENC > 52 (with the exception of *L. hians*). Two subgroups are detectable: group I-a comprises all the Pectinida and Ostreoidea species (Pteriomorphia), two Anomalodesmata (*M. anomiodes* and *L. elliptica*), two Protobranchia (*E. tenuis* and *S. velum*) and the Imparidentia *G. turtoni*, *M. arenaria* and *S. constricta*. The species pertaining to this subgroup show a weak CUB, with ENC 54-58 and $GC_3$ very close to 50% (averaging ~49%).

The subgroup I-b comprises Unionida and Trigoniida (Palaeoheterodonta), *Pinctada* spp. (Pteriomorphia, Pterioidea), eight unrelated Imparidentia and the three Archi-

**Table 1 Effective number of codons (ENC) values in bivalves, ordered from the least to the most biased species.**

| Species | Taxonomic classification[a] | | ENC |
|---|---|---|---|
| *Galeomma turtoni* | Imparidentia | Galeommatoidea | 56.80 |
| *Mya arenaria* | Imparidentia | Myidae | 56.76 |
| *Myochama anomioides* | Anomalodesmata | Cleidothaeridae | 56.52 |
| *Placopecten magellanicus* | Pteriomorphia | Pectinida | 56.44 |
| *Mizuhopecten yessoensis* | Pteriomorphia | Pectinida | 56.25 |
| *Azumapecten farreri* | Pteriomorphia | Pectinida | 56.10 |
| *Ennucula tenuis* | Protobranchia | Nuculoidea | 55.98 |
| *Solemya velum* | Protobranchia | Solemyoidea | 55.73 |
| *Laternula elliptica* | Anomalodesmata | Cleidothaeridae | 55.72 |
| *Argopecten irradians* | Pteriomorphia | Pectinida | 55.65 |
| *Ostrea lurida* | Pteriomorphia | Ostreoidea | 55.61 |
| *Pecten maximus* | Pteriomorphia | Pectinida | 55.61 |
| *Sinonovacula constricta* | Imparidentia | Adapedonta | 55.51 |
| *Mimachlamys nobilis* | Pteriomorphia | Pectinida | 55.47 |
| *Crassostrea virginica* | Pteriomorphia | Ostreoidea | 55.33 |
| *Ostreola stentina* | Pteriomorphia | Ostreoidea | 55.25 |
| *Crassostrea gigas* | Pteriomorphia | Ostreoidea | 55.24 |
| *Crassostrea angulata* | Pteriomorphia | Ostreoidea | 55.13 |
| *Glossus humanus* | Imparidentia | Venerida | 55.04 |
| *Crassostrea corteziensis* | Pteriomorphia | Ostreoidea | 54.95 |
| *Pinctada fucata* | Pteriomorphia | Pterioidea | 54.90 |
| *Crassostrea hongkongensis* | Pteriomorphia | Ostreoidea | 54.88 |
| *Polymesoda caroliniana* | Imparidentia | Venerida | 54.71 |
| *Ostrea edulis* | Pteriomorphia | Ostreoidea | 54.70 |
| *Ostrea chilensis* | Pteriomorphia | Ostreoidea | 54.46 |
| *Saccostrea glomerata* | Pteriomorphia | Ostreoidea | 54.46 |
| *Astarte sulcata* | Archiheterodonta | Crassatelloidea | 53.95 |
| *Pinctada martensi* | Pteriomorphia | Pterioidea | 53.89 |
| *Corbicula fluminea* | Imparidentia | Venerida | 53.88 |
| *Pinctada maxima* | Pteriomorphia | Pterioidea | 53.47 |
| *Hiatella arctica* | Imparidentia | Adapedonta | 53.42 |
| *Arctica islandica* | Imparidentia | Venerida | 53.42 |
| *Cardites antiquata* | Archiheterodonta | Carditoidea | 52.92 |
| *Neotrigonia margaritacea* | Palaeoheterodonta | Trigoniida | 52.87 |
| *Villosa lienosa* | Palaeoheterodonta | Unionida | 52.76 |
| *Margaritifera margatifera* | Palaeoheterodonta | Unionida | 52.68 |
| *Eucrassatella cumingii* | Archiheterodonta | Crassatelloidea | 52.66 |
| *Ellipto complanata* | Palaeoheterodonta | Unionida | 52.64 |
| *Uniomerus tetralasmus* | Palaeoheterodonta | Unionida | 52.62 |
| *Cyrenoida floridana* | Imparidentia | Venerida | 52.35 |
| *Pyganodon grandis* | Palaeoheterodonta | Unionida | 52.32 |

Table 1 (*continued*)

| Species | Taxonomic classification[a] | | ENC |
|---|---|---|---|
| *Lampsilis cardium* | Palaeoheterodonta | Unionida | 52.29 |
| *Donacilla cornea* | Imparidentia | Mactroidea | 52.18 |
| *Meretrix meretrix* | Imparidentia | Venerida | 51.50 |
| *Lamychaena hians* | Imparidentia | Gastrochaenidae | 51.07 |
| *Sphaerium nucleus* | Imparidentia | Sphaeriidae | 51.01 |
| *Mercenaria campechiensis* | Imparidentia | Venerida | 50.98 |
| *Cerastoderma edule* | Imparidentia | Venerida | 50.92 |
| *Ruditapes decussatus* | Imparidentia | Venerida | 50.21 |
| *Ruditapes philipinarum* | Imparidentia | Venerida | 50.13 |
| *Atrina rigida* | Pteriomorphia | Pinnoidea | 49.62 |
| *Diplodonta sp.* | Imparidentia | Cyamiidae | 49.40 |
| *Geukensia demissa* | Pteriomorphia | Mytilida | 47.42 |
| *Cycladicama cumingii* | Imparidentia | Cyamiidae | 47.25 |
| *Perna viridis* | Pteriomorphia | Mytilida | 47.21 |
| *Bathymodiolus azoricus* | Pteriomorphia | Mytilida | 46.65 |
| *Anadara trapezia* | Pteriomorphia | Arcida | 45.72 |
| *Tegillarca granosa* | Pteriomorphia | Arcida | 45.69 |
| *Mytilus galloprovincialis* | Pteriomorphia | Mytilida | 45.58 |
| *Bathymodiolus platifrons* | Pteriomorphia | Mytilida | 45.46 |
| *Mytilus edulis* | Pteriomorphia | Mytilida | 45.34 |
| *Mytilus trossulus* | Pteriomorphia | Mytilida | 45.26 |
| *Mytilus californianus* | Pteriomorphia | Mytilida | 45.02 |
| *Mactra chinensis* | Imparidentia | Mactroidea | 40.65 |

**Notes.**
[a] Based on the revised classification of bivalves by *Bieler et al. (2014)*.

heterodonta species. Compared to subgroup I-a, the observed CUB is slightly higher, including species with ENC ranging from ∼52 to ∼54, but GC$_3$ is remarkably lower than group Ia, averaging ∼43%.

(II) The second major group comprises the remaining 20 species, including the Pteriomorphia groups Mytilida, Arcida and Pinnoidea (represented by the lone species *A. rigida*), together with various Imparidentia species classified as Cardioidea, Cyamiidae, Mactroidea, Sphaeriidae and Venerida, implying that this well-defined group of bivalves with high CUB (ENC ∼40–52) and low GC$_3$ (30–40%, with the outlier *M. chinensis* reaching ∼23%) comprises phylogenetically distant species.

## Codon usage is biased towards A/T ending codons in Bivalvia

Here we report the most commonly used codon(s) for each amino acid, designing as "preferred codons" all the codons used more frequently than expected (RSCU > 1), and we investigate the correlation between their frequency and overall codon usage bias (negative ENC) across the bivalve species analyzed (*Vicario, Moriyama & Powell, 2007*).

Figure 3A summarizes the number of bivalve species where a given codon is preferred, evidencing that, despite the difference in overall ENC values, several preferred and avoided codons are shared by most, if not by all, bivalves: this is the case, for example, of ACA (Thr),

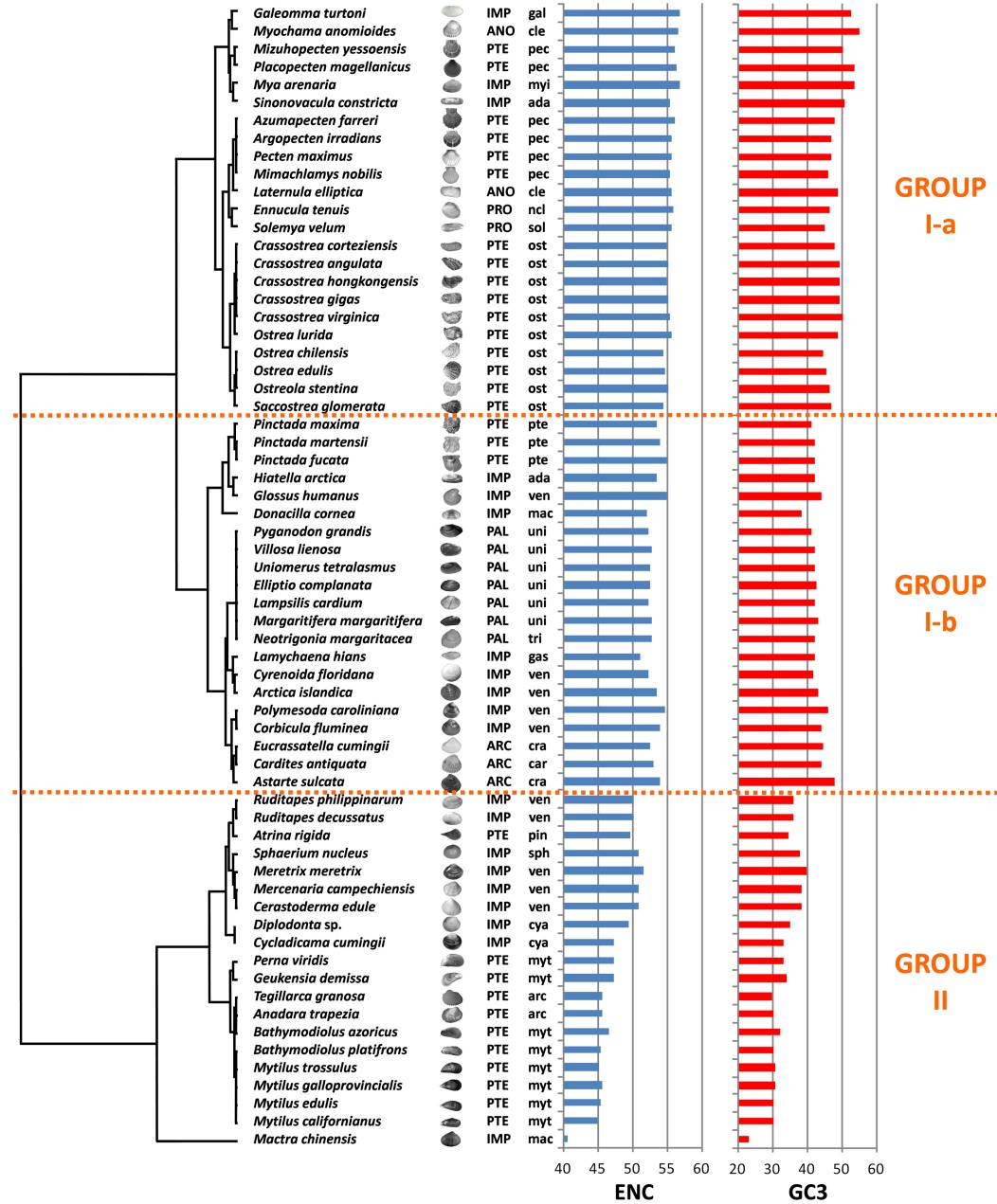

**Figure 2 Clustering of bivalve species according to the variation of codon usage.** The dendrogram was inferred with Cluster 3 by hierarchical clustering, using Euclidean distance as a similarity metric and an average linkage clustering method. Effective number of codons (ENC) and GC$_3$ metric for each species are also displayed. A three-letter code near the species name indicates the taxonomical classification according to *Bieler et al. (2014)*. In detail, capital letters identify one of the six major evolutionary lineages and lowercase letters identify the order. ARC, Archiheterodonta; ANO, Anomalodesmata; IMP, Imparidentia; PAL, Palaeoheterodonta; PRO, Protobranchia; PTE, Pteriomorphia; ada, Adapedonta; arc, Arcida; car, Carditoidea; cle, Cleidothaeridae; cra, Crassatelloidea; cya, Cyamioidea; gal, Galeommatoidea; gas, Gastrochaenidae; mac, Mactroidea; myi, Myida; myt, Mytilida; ncl, Nuculoidea; pec, Pectinida; pin, Pinnoidea; pte, Pterioidea; ost, Ostreoidea; sol, Solemyoidea; sph, Sphaerioidea; tri, Trigoniida; uni, Unionida; ven, Veneroidea.

## A.

| preferred | A | T | C | G |
|---|---|---|---|---|
| AA | 54 Lys | 48 Asn | 16 Asn | 10 Lys |
| AT | 28 Ile | 58 Ile | 19 Ile | / Met |
| AC | 64 Thr | 25 Thr | 9 Thr | 0 Thr |
| AG | 64 Arg | 64 Ser | 0 Ser | 44 Arg |
| TA | / STOP | 39 Tyr | 25 Tyr | / STOP |
| TT | 28 Leu | 59 Phe | 5 Phe | 52 Leu |
| TC | 64 Ser | 60 Ser | 17 Ser | 0 Ser |
| TG | / STOP | 64 Cys | 0 Cys | / Trp |
| CA | 4 Gln | 54 His | 10 His | 60 Gln |
| CT | 0 Leu | 19 Leu | 2 Leu | 49 Leu |
| CC | 64 Pro | 57 Pro | 7 Pro | 0 Pro |
| CG | 3 Arg | 4 Arg | 0 Arg | 0 Arg |
| GA | 49 Glu | 60 Asp | 4 Asp | 14 Glu |
| GT | 21 Val | 42 Val | 1 Val | 52 Val |
| GC | 63 Ala | 64 Ala | 35 Ala | 0 Ala |
| GG | 64 Gly | 52 Gly | 2 Gly | 0 Gly |

## B.

| - ENC | A | T | C | G |
|---|---|---|---|---|
| AA | 0,86 Lys | 0,9 Asn | -0,9 Asn | -0,86 Lys |
| AT | 0,82 Ile | 0,34 Ile | -0,92 Ile | / Met |
| AC | 0,84 Thr | 0,71 Thr | -0,87 Thr | -0,77 Thr |
| AG | 0,95 Arg | 0,9 Ser | -0,89 Ser | -0,59 Arg |
| TA | / STOP | 0,94 Tyr | -0,94 Tyr | / STOP |
| TT | 0,93 Leu | 0,91 Phe | -0,91 Phe | 0,24 NS Leu |
| TC | 0,91 Ser | 0,84 Ser | -0,88 Ser | -0,76 Ser |
| TG | / STOP | 0,75 Cys | -0,75 Cys | / Trp |
| CA | 0,85 Gln | 0,91 His | -0,91 His | -0,85 Gln |
| CT | 0,1 NS Leu | 0,2 NS Leu | -0,92 Leu | -0,91 Leu |
| CC | 0,79 Pro | 0,68 Pro | -0,87 Pro | -0,73 Pro |
| CG | -0,79 Arg | -0,11 NS Arg | -0,88 Arg | -0,87 Arg |
| GA | 0,93 Glu | 0,92 Asp | -0,92 Asp | -0,93 Glu |
| GT | 0,85 Val | 0,9 Val | -0,86 Val | -0,93 Val |
| GC | 0,58 Ala | 0,86 Ala | -0,89 Ala | -0,78 Ala |
| GG | 0,54 Gly | 0,75 Gly | -0,86 Gly | -0,83 Gly |

times preferred

0    32    64

correlation

-1    0    +1

**Figure 3 Codon usage bias in bivalves in mainly due to A/T-ending codons.** (A) Number of bivalve species (out of the 64 selected for this study) where a given codon was preferred (RSCU > 1). (B) Paerson correlation coefficient between the frequency of each codon and overall species CUB (negative ENC); NS, non-significant correlation, based on $F$-test of linear regression.

AGA and AGT (Arg), TCA (Ser), TGT (Cys), CCA (Pro), GCT (Ala) and GGA (Gly). On the other hand, the RSCU values of ACG (Thr), AGC (Ser), TCG and CTA (Leu), TGC (Cys), CCG (Pro), CGC and CGG (Arg), GCG (Val) and GGG (Gly) are always lower than 1, indicating that these codons are avoided in all species. In general, A/T-ending codons appear to be preferred over those ending in G/T, but some notable exceptions exist, including the two C-starting codons encoding the six-fold degenerate amino acids Ser and Arg. While G-ending codons are not uncommon, C-ending codons are almost invariably avoided.

However, when the correlation between codon frequencies and overall CUB of bivalve species is taken into account, the important weight of A/T ending codons on bivalve codon bias becomes evident, (Fig. 3B). Indeed, the high CUB of all the species pertaining to clustering group II (Fig. 2) appears to be mostly resulting from an increased use of A/T-ending codons over those ending in G/C, also explaining the significant negative

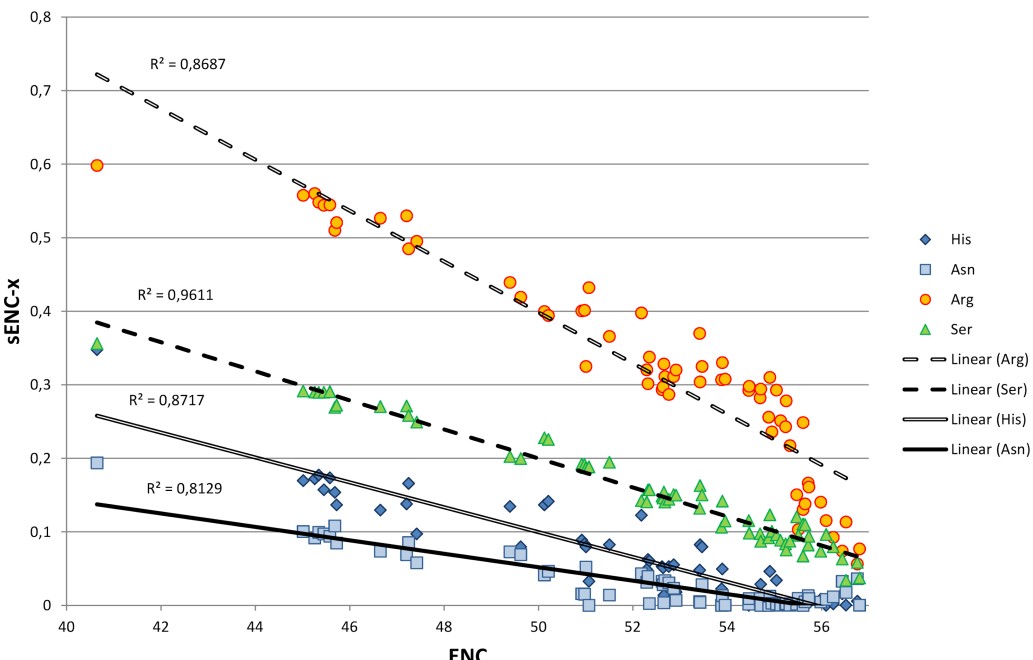

**Figure 4 Simple linear regression analysis exemplifying the different contribution of four amino acids (Asn, Arg, Ser and His) to synonymous codon usage bias.** ENC values for each species are plotted on the *X* axis and represent a measure of synonymous codon usage (lower ENC values indicate a stronger CUB). sENC-x values are plotted on the *Y* axis and represent the relative intensity of CUB for each amino acid in each species. *R* squared correlation values are shown for each regression line. Detailed data for all amino acids are reported in Table S5.

correlation between ENC and $GC_3$ across species (see 'Effects of mutational bias and selection on CUB in Bivalvia' below). This correlation is significant for most codons with, once again, the exception of the C-starting codons encoding the six-fold degenerate amino acids Leu and Arg. On the contrary, the frequency of G/C ending codons is significantly and negatively correlated with CUB in all cases, with the exception of TTG (Leu).

We explored in detail the contribution of different amino acids to CUB in bivalves by calculating s-ENCx values for each amino acid and correlating this parameter to the overall CUB of each species. This measure is a variation of ENC (*Wright, 1990*) which is scaled in a range from 0 to 1 for each amino acid independently from the level of redundancy, and which can be used to estimate the relative intensity of CUB across the 18 degenerate amino acids (*Moriyama & Powell, 1997*). As reported in Table S5 and Fig. 4, the sENC-x values of all amino acids negatively correlate with ENC with significant *p*-values, including those with relatively low average s-ENCx values. The only exception is represented by Gln, which is likely related to the fact that it is the only two-fold degenerate amino acid to display a strong preference for a G/C-ending (CAG) over an A/T-ending codon (CAA) (see Fig. 3A). Overall, Arg is certainly the amino acid which accounts for the greatest CUB in bivalves, as highlighted by the high average s-ENCx value (0.32), followed by Pro, Thr, Cys, Ala, Gly, Ser and Leu, all characterized by values >0.1.

## Effects of mutational bias and selection on CUB in Bivalvia

Although the knowledge of the factors underlying the preferential use of certain codons has remarkably increased over the past few decades, the real contribution of the several potentially contrasting forces involved in shaping CUB still remains a matter of debate in the scientific community.

Mutational bias, intended as the non-randomness of mutational patterns, certainly has a major role in determining codon usage in prokariotes, as well as in many eukaryotes (*Muto & Osawa, 1987*; *Sémon, Lobry & Duret, 2006*). A critical parameter tightly linked to mutational bias is genomic GC content, which in turn strongly influences the coding GC content (*Knight, Freeland & Landweber, 2001*) and $GC_3$ (*Mitreva et al., 2006*). Actually, all amino acids, with the exception of the non-degenerate methionine and tryptophan, tolerate A/T to C/G changes in the third codon position and thus, in organisms where mutational bias is dominant over other factors, the occurrence of G/C-ending codons should follow the total genomic GC content. However, even in such cases some deviations are observed, in particular for the 6-fold degenerated Arg and Leu, which also tolerate synonymous mutations at the first codon position (*Palidwor, Perkins & Xia, 2010*).

In bivalves, average $GC_3$ appears to be tightly related to ENC (Fig. 5), as these two variables could be strongly correlated by linear regression with $R^2 = 0.92$ (*p*-value $= 2.86 \times 10^{-36}$). Overall, while $GC_3$ can assume values as low as ∼23 in *M. chinensis*, $GC_3$ and $AT_3$ contents become even in species displaying weak CUB (and high ENC). Based on these observations and the well-documented correlation between genomic GC content and coding $GC_3$, one could hypothesize that the bivalves pertaining to the clustering group II possess a similar genomic GC content, lower than all the bivalve species included in the heterogeneous clustering group I, thus explaining their common placement in a well-distinct cluster regardless of the reported phylogenetic distance (Fig. 2). On the contrary, the bivalve transcriptomes of the clustering group Ia show a $GC_3$ not far from 50%, accompanied by ENC values higher than 54 (Fig. 5), which would suggest that the absence of a strong CUB is linked to a lack of mutational bias.

However, the genomic data currently available for bivalves do not support this view, since the differences in genomic GC content among species are minimal. The fully sequenced genomes of *P. fucata* and *C. gigas* possess a relatively low G/C composition (33.69% and 32.33%, respectively) (*Takeuchi et al., 2012*; *Zhang et al., 2014*) and partial data from *A. farreri* and *M. galloprovincialis* indicate a similar GC content for these two species (35.75% and 31.65%, respectively) (*Zhao et al., 2012*; *Nguyen, Hayes & Ingram, 2014*). Therefore, while the A/T-rich nature of most bivalve genomes is consistent with a role of mutational bias towards the preferential choice of A/T-ending codons in bivalves ('Codon usage is biased towards A/T ending codons in Bivalvia'), other factors might be taken into account to explain the differences in CUB among these marine organisms.

Among these, selection for translational speed and accuracy are certainly among the most relevant, as they can potentially overcome and mask the effects of mutational bias on CUB. This is the case, for example, of *Drosophila* spp. and of *Strongylocentrotus purpuratus*, which preferentially use G/C ending codons despite having A/T rich genomes (*Vicario,*

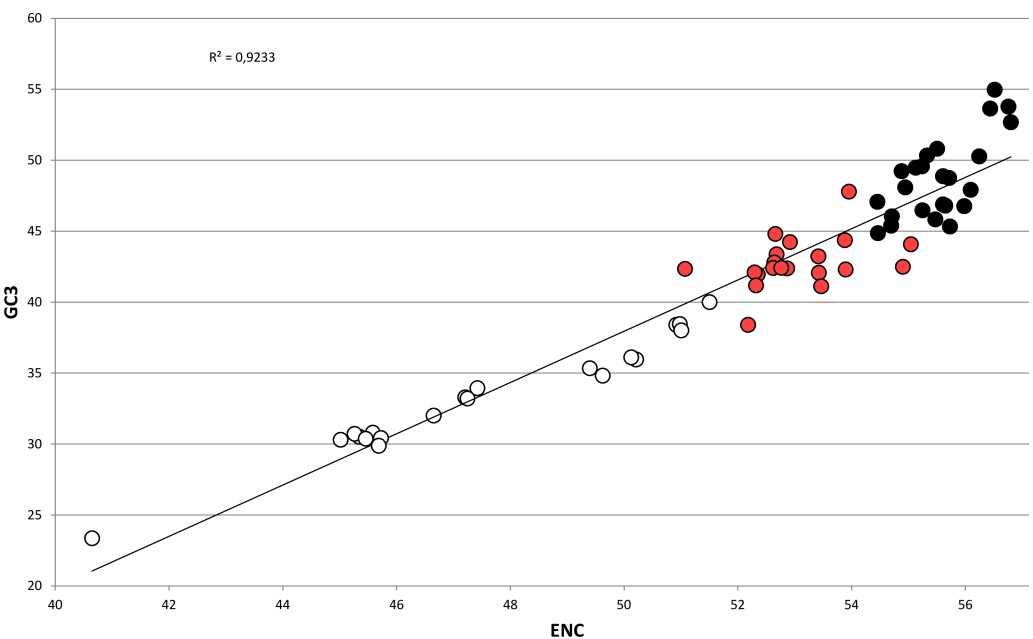

**Figure 5 Regression line defining the correlation between ENC and GC₃ in bivalve species.** Species pertaining to the bivalve clustering group Ia, Ib and II (see Fig. 2) are marked as black, red and white circles, respectively. The $p$-value of the $F$-test of linear regression is $2.86 \times 10^{-36}$.

*Moriyama & Powell, 2007*; *Kober & Pogson, 2013*). The selection for translational speed is evident in many unicellular and multicellular organisms, where the preferential use of optimal codons by highly expressed genes, with the aim to maximize the rate of elongation during protein synthesis (*Marais & Duret, 2001*), has been clearly demonstrated (*Gouy & Gautier, 1982*; *Powell & Moriyama, 1997*; *Duret, 2000*). Nevertheless, the correlation between gene expression and CUB is not universally applicable, as the selection for translational speed appears to be weak in some species (*Hiraoka et al., 2009*).

Natural selection also potentially shapes codon usage to improve translational accuracy, reducing the risk of missense errors during the translational process (*Akashi, 1994*; *Eyre-Walker, 1996*). Even though this model of selection predicts a higher CUB in genes encoding longer proteins, a positive correlation between CUB and protein length has only been observed in bacteria, while on the opposite an unexpected negative correlation has been described in a number of eukaryotes, including *S. cerevisiae*, *D. melanogaster* and *C. elegans* (*Moriyama & Powell, 1998*; *Duret & Mouchiroud, 1999*). Even though differences in selective constraints between prokaryotes and eukaryotes have been evoked to explain this contradiction, the relationship between CUB and protein length still remains obscure (*Marais & Duret, 2001*).

The investigation of these two selective forces in bivalves is complicated by the incomplete/fragmented nature of *de novo* assembled transcriptomes and by the limited gene expression resources available. We correlated CUB with gene expression (in three tissues: digestive gland, gills and hemocytes) and with ORF (protein) length in *C. gigas*, the most appropriate bivalve species for this purpose due to the completeness of its

**Table 2 Influence of mutational bias and selection on codon usage bias in *Crassostrea gigas* and *Mytilus galloprovincialis*.** Paerson correlation coefficients and $p$-values of $F$-test for linear regression analysis are shown.

| | *Crassostrea gigas* | *Mytilus galloprovincialis* |
|---|---|---|
| Coding GC$_3$ | 49.27% | 30.80% |
| Global ENC | 55.24 | 45.58 |
| Genomic GC content | 33.69% | 31.65% |
| Mutational bias | Towards A/T-ending codons | Towards A/T-ending codons |
| Correlation between CUB and protein length | 0.03 ($p$-value $9.14 \times 10^{-8}$) | 0.09 ($p$-value $3.75 \times 10^{-18}$) |
| Correlation between GC$_3$ and protein length | 0.05 ($p$-value $9.69 \times 10^{-18}$) | $-0.10$ ($p$-value $5.43 \times 10^{-22}$) |
| Selection for translational accuracy | Towards G/C-ending codons | Towards A/T-ending codons |
| Correlation between CUB and gene expression (hemocytes) | 0.04 ($p$-value $8.99 \times 10^{-12}$) | 0 (NS) |
| Correlation between GC$_3$ and gene expression (hemocytes) | 0.03 ($p$-value $1.44 \times 10^{-9}$) | 0.07 ($p$-value $3.99 \times 10^{-11}$) |
| Correlation between CUB and gene expression (digestive gland) | 0.05 ($p$-value $1.16 \times 10^{-17}$) | 0 (NS) |
| Correlation between GC$_3$ and gene expression (digestive gland) | 0.06 ($p$-value $1.04 \times 10^{-20}$) | 0.11 ($p$-value $2.67 \times 10^{-24}$) |
| Correlation between CUB and gene expression (gills) | 0.07 ($p$-value $4.20 \times 10^{-29}$) | 0 (NS) |
| Correlation between GC$_3$ and gene expression (gills) | 0.06 ($p$-value $6.14 \times 10^{-23}$) | 0.12 ($p$-value $1.04 \times 10^{-28}$) |
| Selection for translational speed | Towards G/C-ending codons | Towards G/C-ending codons |
| Correlation between CUB and GC3 | $-0.16$ ($p$-value $5.42 \times 10^{-148}$) | $-0.53$ ($p$-value 0) |
| Prevailing factor at the whole protein-coding transcriptome scale | Mutational bias | Mutational bias and selection for translational accuracy |

genome annotation and availability of gene expression data. We observed a significant positive correlation between CUB (negative ENC) and gene expression in the three tissues analyzed, as well as between CUB and ORF (protein) length (Table 2), which would suggest that both selection for translational speed and accuracy are actively shaping codon usage in oyster. However, we also observed a highly significant, negative correlation between GC$_3$ and gene expression and between GC$_3$ and protein length, which seem to contradict the mutational bias given by the A/T-rich nature of the oyster genome. This observation matches the results obtained in a previous work conducted with limited expression data based on Sanger EST sequencing, which suggested that translational selection acts as a contrasting force to mutational bias in oyster, effectively counteracting its action in highly expressed genes (*Sauvage et al., 2007*). Our data further indicate that, besides the selection for translational speed, also the selection for translational accuracy provides a contribution to the selection of G/C-ending codons in oyster.

The contrasting action of mutational bias and selection becomes particularly evident while taking into consideration the correlation between CAI, a directional measure of CUB which is based on a reference set of highly expressed genes (*Sharp & Li, 1987*), and ENC, which on the other hand is a non-directional measure which does not permit to appreciate the contribution of opposite forces (in this case mutational bias towards A/T-ending codons and selection towards G/C-ending codons). Indeed, in oyster and in all the other species clusterized in group I (see Fig. 2), the scatter in the correlation plot between CAI and ENC appears to be quite relevant (Fig. 6; Paerson correlation coefficients are $-0.44$ for *C. gigas* and $-0.38$ for *P. magellanicus*). This indicates that the mutational

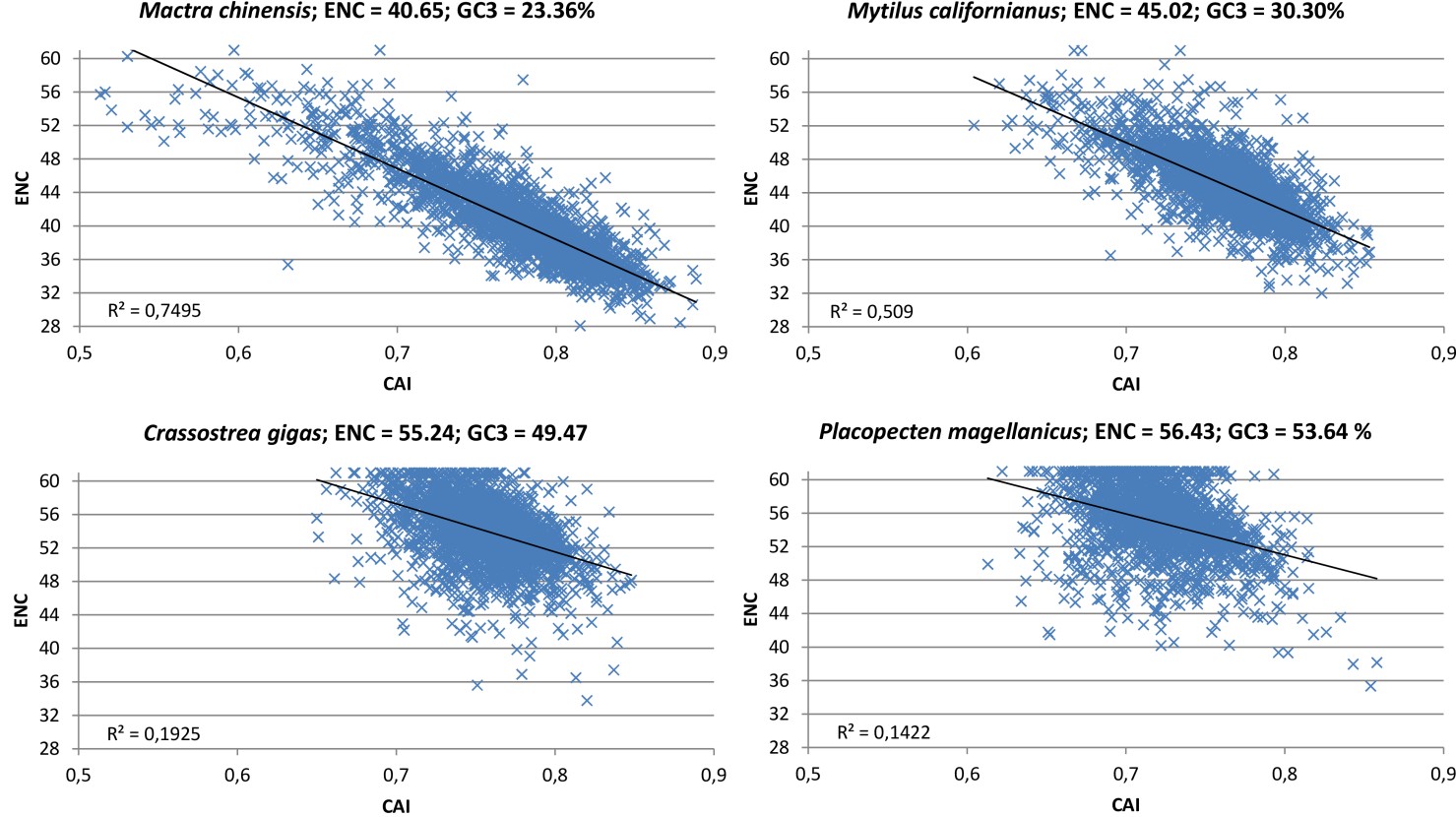

**Figure 6 CAI vs. ENC plot.** Scatter plot of CAI (*X* axis) vs. ENC (*Y* axis) for four representative bivalve species: *Mactra chinensis*, *Mytilus californianus*, *Crassostrea gigas* and *Placopecten magellanicus*. The reference set of highly expressed genes for each species is based on the orthologous genes of *C. gigas* (see 'Materials and Methods').

bias towards A/T is counterbalanced by natural selection in favor of G/C-ending codons in a relevant number of genes. However, the significant correlation between ENC and $GC_3$ at the whole protein-coding transcriptome level (Paerson correlation $= 0.16$, *p*-value $= 5.42 \times 10^{-148}$) indicates that the weight of A/T mutational bias is still dominating over that of G/C selection in most oyster genes.

Overall, it is likely that in all species pertaining to clustering group I, whose coding $GC_3$ sensibly deviates from the frequency expected by genomic GC content, the mutational bias towards A/T-ending codons is countered by the opposite forces of selection for translational speed and accuracy, leading to moderate/low CUB.

On the other hand, the correlation between CAI and ENC in the bivalve species pertaining to clustering group II is highly significant (Fig. 6, Paerson correlation coefficients are $-0.87$ for *M. chinensis* and $-0.71$ for *M. californianus*). To better interpret this result, we extended the analysis performed in oyster to a representative species of this group, *M. galloprovincialis*, limiting our calculations to full-length protein-coding transcripts (Table 2). Overall, like in oyster, the significant positive correlation between gene expression and $GC_3$ over three different tissues indicates that selection for translational

speed acts towards G/C-ending codons. However, unlike oyster, this appears to have no effect on CUB on the whole transcriptome scale, as the correlation between gene expression and ENC was not significant. In addition, we observed that, in contrast with oyster, the selection for translational accuracy is biased towards A/T-ending codons (i.e., $GC_3$ and protein length are negatively correlated). Overall, the *M. galloprovincialis* average $GC_3$ is 30.80%, a value which is very close to the GC genomic content, further indicating that mutational bias and selection for translational accuracy, which both act in favor of A/T-ending codons, are largely dominating over the opposite selection for translational speed in mussel.

## CONCLUSION

In this paper we presented a comprehensive and well supported overview of CUB in 64 different bivalve species, including both marine and freshwater species, based on the analysis of nearly 3,000 evolutionarily conserved genes. To the best of our knowledge, this is the first time that CUB is systematically investigated in such a large and important class of invertebrates, the largest one after Insecta. We show that bivalves can be divided into two distinct groups, based on the intensity of the bias towards the use of optimal codons. While in many species CUB appears to be relatively weak, Mytilida, Arcida and several Imparidentia species show a remarkable preference for A/T-ending codons.

Given the poor correlation between CUB and bivalve taxonomy, other factors are expected to drive the evolution of CUB towards the same direction in distantly related species. We investigated this issue in *C. gigas* and *M. galloprovincialis*, two species displaying low and high CUB, respectively. Our analyses pointed out that:

(i) Bivalves are subject to mutational bias towards A/T-ending codons, due to the low GC content of their genomes. Although the nucleotide genome composition should be reflected by a low $GC_3$, some species display an almost even nucleotide content at the third codon position, which could be explained by the contrasting action exerted by other factors.

(ii) Gene expression is positively correlated with $GC_3$, which indicates a selection towards G/C-ending codons, acting to maximize the rate of elongation of protein synthesis for highly expressed transcripts. The different intensity of this selective force, opposed to mutational bias, might partly explain differences in CUB among species with similar genomic GC content.

(iii) Although the selection for translational accuracy, evidenced by the correlation between protein length and CUB, is also active in bivalves, the direction of this selective force varies across species, either reinforcing (in *M. galloprovincialis*) or counteracting (in *C. gigas*) mutational bias.

In conclusion, multiple factors contribute to CUB in bivalves, and the specific weight of each of them is still difficult to be determined with certainty considering the limited genomic resources available for most species. However, based on the data collected so far, the different intensity of the opposing forces represented by mutational bias towards

A/T-ending codons and by selection for translational speed towards G/C-ending codons appear to be two major players in this process. Further study will be certainly needed to ascertain whether this model can be extended to all bivalve species once additional genomic resources will become available. Such analyses could also benefit from the integration of additional data such as the abundance of isoaccepting tRNAs and data concerning effective population size of each species.

This large scale analysis supports the progressive understanding of molecular genome evolution in bivalves and it is potentially useful for many different applications. For example, as we have previously explained, codon usage is known to widely vary across genes based on their expression level. The calculation of CAI could be used to predict the expression level and the transcriptional efficiency of unknown genes and to assess the adaptation of viral genes to their bivalve hosts (*Sharp & Li, 1987*). Codon bias also has profound implications on codon-based phylogenetic reconstructions, which could be optimized to avoid the over-estimation of divergence between distantly related species and the under-estimation of divergence between closely related ones (*Christianson, 2005*).

But, above all, one of the key applications of codon usage in the post-genomic era is certainly the annotation of newly sequenced genomes. As the raw genomic sequences of new organisms become available, new tools are required to efficiently identify genes and to predict their structure and boundaries, in particular in non-model invertebrate species, which show a very high proportion of genes without similarity to sequences deposited in public databases. The oyster genome has already evidenced that homology-based annotation methods perform rather poorly for certain gene families, highlighting the need for integrating additional parameters in gene prediction algorithms to optimize annotation (*Gerdol, Venier & Pallavicini, 2014*). A number of algorithms which take into account species-specific CUB information have been developed, trying to identify protein coding sequences based on their congruence with a reference codon usage table (*Gribskov, Devereux & Burgess, 1984*). This task is particularly important and complicated in Bivalvia, since genomic analyses in this taxa are quickly expanding and only two species present a completely sequenced and annotated genome (*Suárez-Ulloa et al., 2013*).

**Abbreviations**

| | |
|---|---|
| **CAI** | Codon Adaptation Index |
| **CDS** | Coding sequence |
| **CUB** | Codon Usage Bias |
| **ENC** | Effective Number of Codons |
| **GC$_3$** | GC content at the third codon base |
| **NGS** | Next Generation Sequencing |
| **ORF** | Open Reading Frame |
| **RNA-seq** | RNA-sequencing |
| **RSCU** | Relative Synonymous Codon Usage |

### Funding

This work was supported by BIVALIFE (FP7-KBBE-2010-4). The funders had no role in study design, data collection and analysis, decision to publish, or preparation of the manuscript.

### Grant Disclosures

The following grant information was disclosed by the authors:
BIVALIFE: FP7-KBBE-2010-4.

### Competing Interests

The authors declare there are no competing interests.

### Author Contributions

- Marco Gerdol conceived and designed the experiments, performed the experiments, analyzed the data, contributed reagents/materials/analysis tools, wrote the paper, prepared figures and/or tables, reviewed drafts of the paper.
- Gianluca De Moro analyzed the data, contributed reagents/materials/analysis tools.
- Paola Venier analyzed the data, reviewed drafts of the paper.
- Alberto Pallavicini analyzed the data, contributed reagents/materials/analysis tools, reviewed drafts of the paper.

### Data Availability

The research in this article did not generate any raw data.

### Supplemental Information

Supplemental information for this article can be found online at http://dx.doi.org/10.7717/peerj.1520#supplemental-information.

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
