# Peer review of "Analysis of synonymous codon usage patterns in sixty-four different bivalve species"

_PeerJ, doi:10.7717/peerj.1520_

## Round 0.1 · original submission · Major Revisions

The manuscript should be improved and rewritten in a succinct manner avoiding more known descriptions.

·

Basic reporting

No main Comments.
Introduction complete.
The topic is relevant for the genomic research both to help Bivalvia genome annotations and to understand CUB pattern and possibly processes across biodiversity.

Experimental design

The article is mainly descriptive but explore some correlation between variables that are informative on the process that produced the described patterns.
__ORF identification__:
The analysis depends from the assembly of transcriptome and ORF identification. This analysis was done ab initio and ex novo by the authors of the manuscript. Unfortunately the authors gives very scant information on the quality of this process and do not comment on possible bias. The software Transcoder use codon usage of the 500 best ORF( in length, within canonical set [ presence of clear start and stop codon]) to find the rest of the ORFs. This could cause the transcript set to be more homogenous than the real one in term of CUB. Unfortunately no homologous search was implemented as Transcoder would allow. Some discussion on these choices and on the quality of the outcomes could be useful to evaluate results.
Further the few information available (N50==450nt-> 150aa) show a very fragmented reconstruction of the proteomes, given that N50 ( a strange statistics for protein size) gives value larger than median and 150aa is a very small protein (in uniprot only 25% of protein of bivalvia are equal or smaller than this value) .
I would have preferred that comparison on CUB would have been done based well annotated subset of genes in C. gigas and an homologous/paralogous filtering across the 67 genomes. I think in CUB analysis is more relevant to have an unbiased sample more than a more complete sample.
__Preferred Codons__:
pag 10 198-202 the authors make reference to Vicario, Moriyama & Powell, 2007 ) for their method but within that article the RSCU method is not considered optimal and indeed get less crisp results than others method in the 2007 paper. Given that the correlation between overall ENC and codon frequency use the same data than the one in this MS maybe a more clean result could be obtained using correlation.
The sentence "Not surprisingly, most amino acids showing a preference for certain codons did in turn show avoidance for others" is more to do with mathematical property of RSCU than a biological reality. A correlation approach would be more telling allowing to distinguish between no correlation and positive correlation between ENC and codon frequency.

Validity of the findings

The overall analysis assume a substantial homogeneity of CUB and GC across genomes. This assumption makes calculate ENC value not for each ORF/transcript identified but for each group of size, or expression. This is probably a misleading and unnecessary simplification that mix patterns.
In the conclusion the author state "The identification of a reference set of highly expressed housekeeping genes, would allow the calculation of the Codon Adaptation Index (CAI)" but highly expression genes alone even if not bona fide house keeping could be in any case a good set to build CAI index. A very simple CAI and ENC graph could give a synthetic depiction of CUB variation within each of the genomes. The amount of variation of CUB and its relation with expression could help to distinguish process linked to overall genome composition from the one linked to translation, while with the actual analysis the main points are that CUB is mild in Bivalvia and effect of translation are visible only for highly expressed genes.

Additional comments

the supplemental table was not downloadable.

Reviewer 2 ·

Basic reporting

The results are based on methods used in this study. However, I strongly feel that the manuscript is written in highly descriptive manner. Many of the reports made here are already known. Nothing new here, just another organism being studies. I would request the authors to rewrite it in very succinct manner without being descriptive.

Experimental design

Although most part of the methods are logical to the aim of the study, I am not convinced about the methods of inferring selection and phylogeny from codon bias data. This is because:
1. the authors have done nothing to control the effect of mutational bias on codon bias. This is critical because the relationship, without considering mutational bias, codon bias will fluctuate if there is selection on G/C (bias in codon usages) and mutational bias is on A/T positions.
2. The most important issue of this paper is that the authors try to suggest lack of relationship between phylogeny and codon bias without taking into consideration of ancestral genetic recombination in the coding sequences. For example, intracodon recombination (see Arenas M, Posada D. Coalescent simulation of intracodon recombination. Genetics. 2010 Feb;184(2):429-37) can skew the relationship between codon bias and phylogeny is not properly controlled in codon bias studies . Without considering the coalescent events into consideration, current study can have very misleading.

Validity of the findings

Unless the experimental designs are properly controlled as I indicated above, the findings may have pitfalls.

Additional comments

Please rewrite the paper in a concise and precise manner that reveals new findings without describing about lot of things that are already known about codon bias (for example codon bias and its relation with GC3, this is just one example). Present results that is only novel.

---

## Round 0.2 · accepted · Accept

Authors have imrpoved the manuscript according to the proposed suggestions.